# Moringa and Graphite as Additives to Conventional Petroleum-Based Lubricants

**Nadiège Nomède-Martyr** [1,*] **, Philippe Bilas** [1,2] **, Yves Bercion** [2] **and Philippe Thomas** [1]

[1] Groupe de Technologie des Surfaces et Interfaces (GTSI), EA 2432, Faculté des Sciences Exactes et Naturelles, Université des Antilles, CEDEX, 97159 Pointe-a-Pitre, France; philippe.bilas@univ-antilles.fr (P.B.); philippe.thomas@univ-antilles.fr (P.T.)

[2] Centre Commun de Caractérisation des Matériaux des Antilles et de la Guyane, Faculté des Sciences Exactes et Naturelles, Université des Antilles, CEDEX, 97159 Pointe-a-Pitre, France; yves.bercion@univ-antilles.fr

[*] Correspondence: nadiege.nomede-martyr@univ-antilles.fr; Tel.: +590-690-32-7586

**Abstract:** Many researches are focused on the tribological performances of pure vegetable oil in order to replace the conventional mineral engine oils. This work investigates the influence of local moringa oil (noted VO) on the performances of lubricants formed from a blend of dodecane and graphite particles at ambient temperature. In a first part, a reduction of about 50% of friction properties of dodecane is observed when adding small amounts of moringa oil (VO), which is intended to be used as a bio-base performance additive in lubricant formulations. The friction properties of their blends with graphite, generally employed as solid lubricant additive, showed an adsorption effect of fatty acid molecules. The more promising results were obtained for the blend containing 2 w% of VO. Physicochemical characterizations of the tribofilms evidence the good antiwear properties of the lubricant.

**Keywords:** vegetable oil; graphite; additive; mixed lubrication





## 1. Introduction

Green lubrication presents an increasing interest in world industrial and economic development. Indeed, commercial lubricants being petroleum-based are the subject of numerous studies due to the progressive depletion of the world reserves of fossil fuels but also owing to concern on their environmental impact. Conventional lubricants are composed of base oil and additives conferring specific properties to the lubricant. Friction reduction additives are used to ensure the lubricating properties in the friction boundary regime [1–5]. Other additives act on the oiliness of the lubricant [6–9]. Natural oils developed with vegetal or animal biomass, and fats present better friction and wear performances than mineral oils. If the use of vegetable oils as lubricant base oil is not economically possible, the addition of vegetable oils to conventional mineral oils is interesting to improve the tribological performances of the lubricants and to reduce its environmental impact.

Indeed, the amphiphilic properties due to the presence of fatty acids in vegetable oils improves lubrication and antiwear performances compared to mineral or synthetic lubricant oils. Due to their adhesion to metallic surfaces, the long chain of polar fatty acids constituting the structure of the triacylglycerol is responsible for the interest of using natural oils in boundary lubrication by creating a protective thin monolayer, which allows us to reduce friction and wear of the sliding surfaces [10]. Numerous studies have been focused on vegetable oils as surfactant molecules added to engine oils to reduce friction in the boundary lubrication regime [11–14]. In 2018, Bahari et al. experimented with the tribological response of vegetable oils (palm oil, soybean oil) and their blends with mineral engine oil in a reciprocating sliding contact running in severe conditions [15]. The presence of the vegetable oil strongly influences the lubrication performances of the mineral oil/vegetable oil blends. In 2020, Fry et al. studied the adsorption of organic

friction modifier additives (octadecylamine, oleylamine, oleic acid and glycerol) in hexane with a rubbing contact formed by stationary glass ball and a rotating silicon disk under the boundary lubrication regime [16]. They showed the impact of the layer thickness and the surface coverage depending of the molecular structure of organic friction modifier. In 2013, Pereira et al. analyzed the natural biodegradable oils (sunflower oil, high oleic sunflower oil, castor oil and ECO-350 recycled oil) as an alternative to traditional canola oils used for minimum quantity of lubrication [17]. The tribo-rheological performances and investigations of lubricants evidences that high oleic sunflower oil improves tool life and is a feasible alternative to walk towards a total ecofriendly machining process. Moringa oil presents a high oleic concentration, which allows it to improve oxidation stability over many other natural oils [18]. Salaheeden (2014) and Tulashie (2019) have shown that moringa oil is a potential source for bio-fuel due to its high concentration in oleic acid and low concentration of polyunsaturated fatty acids [19,20]. Kerni et al. (in 2019) evaluated the effect of nanoparticles (CuO and hBN) in different concentrations on the friction and wear behavior of epoxidized oil; olive oil consists of 85% unsaturated fatty acids [21]. They observed that the addition of 0.5 w% concentration of nanoparticles in olive oil results in the exhibit minimum friction coefficient.

Many researchers have reported that the addition of graphite nanoparticles can improve the tribological properties of pure oil [9,22–25]. Graphite is well known for its friction properties due to its lamellar structure, in which carbon atoms are strongly bonded (covalent bonds) in graphene sheets, the layers are separated by weak Van der Waals forces. The good friction properties of graphite are due to alignment of graphene layers parallel to the sliding direction [26–28]. In 2015, Su et al. investigated the tribological properties of graphite nanoparticles as LB2000 vegetable-based oil additive with a pin-on-disk friction and wear tester [29]. They show that the smaller particles allow for a lower friction coefficient and reduce the wear volume of the disk.

The objective of this work is to investigate the possibility to use local biomass to improve the performances and reduce the environmental impact of petroleum-based lubricants. The effect of the addition of moringa oil (noted VO) as base additive, and graphite as friction reducer is studied. Dodecane is used as base oil. In the first part, the influence of small amounts of moringa oil (VO) in base oil has been studied. Then, three formulations of lubricants containing different percentages of moringa oil (VO) mixed with a blend of dodecane and fixed weight percentage of graphite particles have been investigated. The second part is focused on the physicochemical characterizations of the lubricant presenting the best friction properties. Infrared spectroscopy and thermogravimetric analyses are performed on a VO/dodecane blend before friction experiments, then Raman spectroscopy and SEM experiments are carried out on a graphite/VO/dodecane blend in order to identify key parameters for friction reduction.

## 2. Materials and Methods

Dodecane ReagentPlus 99% used in this study as base oil was provided by Sigma-Aldrich. Vegetable oil is local moringa oil (VO) extracted by Phytobokaz Laboratory (Guadeloupe, France). The fatty acids composition of moringa oil is presented in Table 1 (industrial analysis of Phytobokaz). VO is mainly composed with monounsaturated fatty acid. Exfoliated graphite particles is used as a solid friction reduction additive (Timcal Society). Graphite particles thickness is about 100 nm with an average size of 40 μm. The ratio between size and thickness is about 400.

Blends containing 0.5, 1, 1.5 and 2 w% of VO in dodecane were prepared. The mixture preparation consists of simply weighing with a precision of 0.01 mg. Three lubricants' compositions were prepared by adding 1 w% of graphite in base oil composed of 1, 2 and 3 w% of VO in dodecane by the same weighing technique. The dispersion of the different blends was obtained in ultrasonically bath during 5 min.

**Table 1.** Composition of moringa oil.

| Fatty Acid Methyl Ester | | % Mole Fraction |
|---|---|---|
| Palmitic | C16:0 | 6.09 |
| Palmitoleic | C16:0 | 1.94 |
| Stearic | C18:0 | 3.77 |
| Oleic | C18:1 | 75.33 |
| Linoleic | C18:2 | 0.90 |
| Linolenic | C18:3 | 0.29 |
| Arachidic | C20:0 | 2.47 |
| Behenic | C22:0 | 5.67 |
| Lignoceric | C24:0 | 1.01 |

The friction properties of materials were measured at room temperature (25 °C) with a reciprocating ball-on-plane tribometer consisting of a AISI 52100 steel ball rubbing against a static AISI 52100 steel plane (Figure 1). The ball with a diameter of 1 cm was brought in contact of the plane with a normal load of 10 N. The alternative motion of the ball was performed with a sliding speed of 4 mm·s$^{-1}$. The frequency is 1 Hz. The tangential force $F_T$ was estimated with a computer-based data acquisition system. The friction coefficient value was calculated as $\mu = \frac{F_T}{F_N}$. Two thousand friction cycles were performed, a cycle corresponding to an alternative motion of the ball. According to Hertz theory, such tribological conditions lead to maximum contact pressure of 1 GPa and a contact diameter of 140 μm. The generation of multidirectional stripes favors the adherence of graphite particles on the sliding surfaces. For all experiments, the initial roughness of the steel ball is about 50 μm. Before friction experiments, both steel materials were successively cleaned in ultrasonic acetone and ethanol baths. A drop of selected mixture was deposed on the plane before the friction experiment. The blends with VO and dodecane are referred to as liquid conditions, and the friction coefficient measured is noted $\mu_{w\%VO+dodecane}$. For mixtures containing graphite particles, the notation is $\mu_{Graphite+w\%VO+dodecane}$.

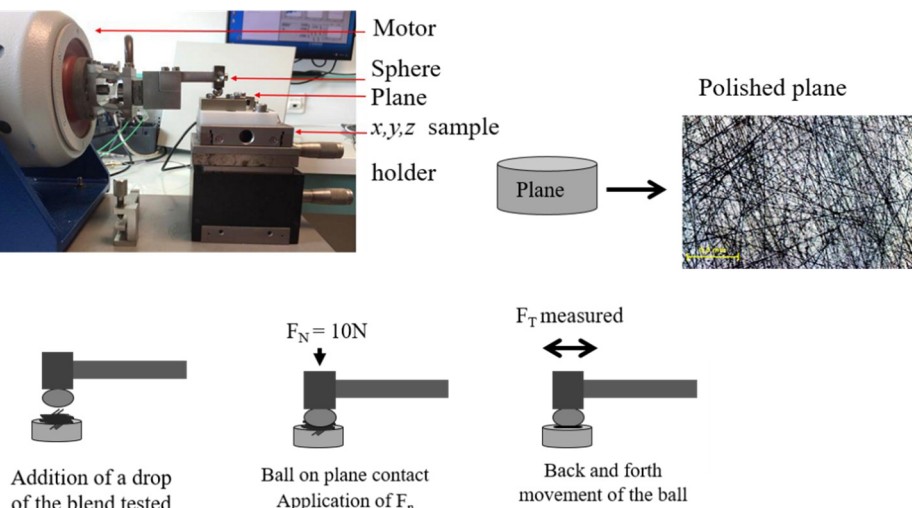

**Figure 1.** Picture of reciprocating ball on plan tribometer with a schematization of friction experiment. SEM image of the multidirectional stripes generated on the steel plane in order to assure the presence of solid particles in the sliding contact.

Fourier transform infrared spectroscopy (FTIR) analyses were performed to identify the functional groups in the blends using a PerkiElmer Spectrum Two spectrometer (Waltham, MA, USA) with a range of 4000 to 50 cm$^{-1}$ wave numbers and a resolution of 4 cm$^{-1}$. Thermogravimetric experiments were carried out with Setaram (Caluire-et-Cuire, France) device under argon at a heating rate of 2 °C/min from room temperature to 700 °C of about 20 mg of sample. The same conditions were used for all the tests. The onset

temperature ($T_{onset}$) and the maximal temperature ($T_{max}$) were reported. The viscosity parameter of the blends without particles was measured by a modular compact rheometer (Anton Paar, Graz, Austria) at ambient temperature with a cone/plane contact. The cone has a diameter of 50 mm and angle of $2°$. The plane has a diameter of 50 mm. The share rate is 0.01 to 1000 $s^{-1}$. Scanning electron microscopy (SEM) using secondary electron imaging characterized the particles and their corresponding tribofilms with a FEI Quanta 250 microscope (Hillsboro, OR, USA). Both samples were analyzed by Raman spectroscopy performed with a HR 800 Horiba multi-channel spectrometer (Kyoto, Japan) using a Peltier-cooled CCD detector for signal recording. The exciting line was 532 nm wavelength line (ND YAG laser). The steel planes were rinsed before Raman analysis in order to eliminate the residual particles.

## 3. Results and Discussion

### 3.1. Influence of the Presence of VO as Bio-Additive

Figure 2 presents the friction coefficient obtained at 1000 cycles, and wear scar diameters were measured on the ball for pure moringa oil (VO) and as a function of the percentage of VO added in dodecane. The comparison of the wear traces diameter to the theorical one (Hertz's theory) allows us to evaluate wear of the ball and in consequence the antiwear properties of the tested lubricants. The friction and wear values obtained for pure dodecane are high; $\mu_{pure\ dodecane} = 0.18 \pm 0.02$ and $\varnothing_{pure\ dodecane} = 280 \pm 10$ μm characterizing severe wear and friction conditions. Contrary to dodecane, the tribological properties of pure VO are weak $\mu_{pure\ VO} = 0.070 \pm 0.005$ and $\varnothing_{pure\ VO} = 150 \pm 10$ μm confirming the excellent properties of vegetable oils. For the VO/dodecane blends, the friction coefficients are not as low as the one obtained for pure VO, but the influence of the presence of VO strongly improves the tribological properties of dodecane. We observe a progressive reduction then a stabilisation of the friction coefficient value according to the percentage of VO added in dodecane. The friction coefficient, $\mu_{0.5w\%VO+dodecane} = 0.13 \pm 0.010$, decreases down to $0.1 \pm 0.005$ from the blend with 1.5 w% of VO. The wear diameter is reduced from 180 to $145 \pm 5$ μm as a function of the w% of VO added. These results lead us to conclude about an excellent tribological influence of the presence of VO. Our results are in good agreement with the literature. Bahari et al. (2018) showed that the presence of vegetable oils improves and dominates the tribological properties of minerale/vegetable oil blends [15]. In 2021, Ponomarenko et al. found that when sunflower oil is added to mineral transmission oil, strong boundary layers are formed during friction reducing wear and friction [12]. At ambient temperature, Reeves et al. (2015) demonstrated that natural oils with high oleic acid concentration present better friction performances [30]. Fry et al. (2020) have demonstrated that the properties of organic friction modifer (oleylamine, oleic acid … ) adsorbed layers govern the friction by forming an adsorbed layer with critical thickness necessary to provide low friction [31]. Our results also suggest a benefical effect of adsorbed film of fatty acid molecules on the friction performances of dodecane.

On the basis of these first results, mixtures containing graphite particles in 1, 2 and 3 w% of VO/dodecane blends were prepared. Figure 3 presents the friction curves obtained for the different mixtures containing 1 w% of graphite in pure VO, in pure dodecane and the three VO/dodecane blends. It can be interresting to note that the friction coefficient value of pure graphite particles is $\mu_{pure\ graphite} = 0.12 \pm 0.01$ in our experimental conditions. All the friction curves decrease during the first cycles down to a stable value after 500 cycles, except for the mixture containing 3 w% of VO. This first part of the friction curves can be attributed to an induction period during in which the tribofilm is built. The curves are then quite stable until the end of the friction experiment. Figure 4 recapitulates the friction coefficient values obtained with the different mixtures containing graphite at 2000 cycles. The friction coefficient value obtained in the presence of pure VO is $\mu_{Graphite+pure\ VO} = 0.09 \pm 0.01$, whereas in the presence of pure dodecane, an important reduction is obtained, $\mu_{Graphite+dodecane} = 0.06 \pm 0.005$, implying a viscosity effect on friction properties of graphite. The friction coefficients of the

graphite/1 w%VO/dodecane and graphite/2 w% VO/dodecane blends are closed to the graphite/dodecane, whereas graphite/3 w% VO/dodecane presents a higher friction coefficient. In previous studies (2021), we have demonstrated the influence of the presence of liquid on the tribological performances of graphite [32]. Indeed, at the addition of dodecane, an immediate and drastic reduction has been evidenced due to simultaneous presence of liquid and particles in the sliding contact. This liquid effect is influenced by its viscosity. An important reduction is observed in the presence of pure dodecane due to weak viscosity $v_{pure\ dodecane} = 1.383$ mPa·s, whereas the reduction is less with pure moringa oil, $v_{pure\ VO} = 87$ mPa·s. However, the viscosity of the different blends containing VO and dodecane are similar $v_{1w\%VO+dodecane} \approx v_{2w\%VO+dodecane} \approx v_{3w\%VO+dodecane} \approx 2.37$ mPa·s when the friction coefficient values are different in the presence of graphite. Consequently, no specific action of viscosity can be supposed. The presence of liquid in the sliding contact is not enough to explain the friction reduction differences. By using the Fry et al. demonstration about critical thickness of adsorbed organic molecules allowing low friction, we can suppose an adsorption effect of fatty acid molecules thickness on the graphite particles and steel surfaces. Our results suggest that the presence of weak amounts of fatty acid molecules in dodecane governs the tribological properties of the blends.

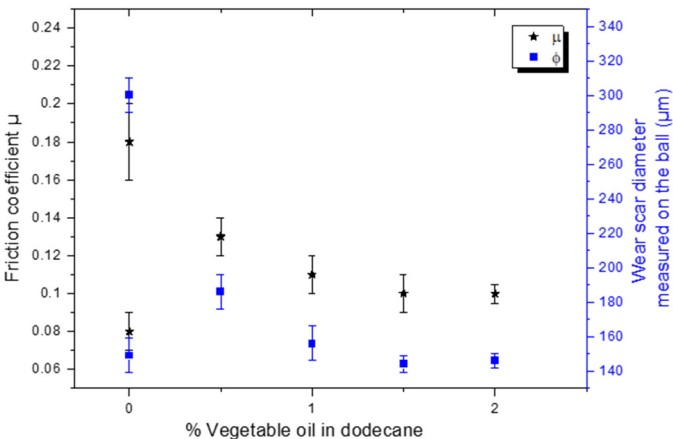

**Figure 2.** Evolution of the friction coefficient (μ) and the wear scar diameter measured on the steel ball (Ø) for pure VO and as a function of the weight percentage of VO added in dodecane blends.

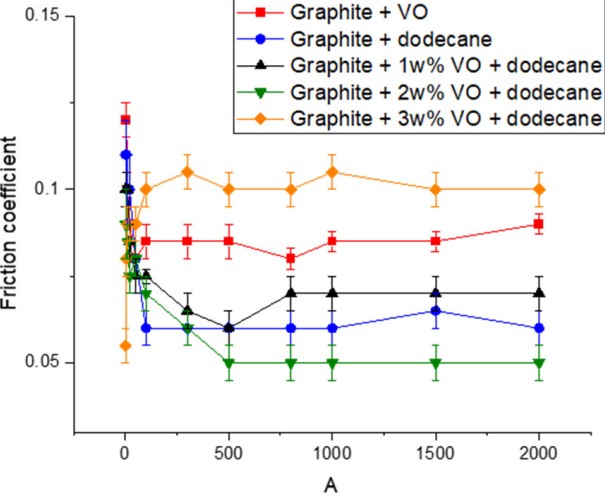

**Figure 3.** Friction curves of the mixtures containing 1 w% of graphite particles.

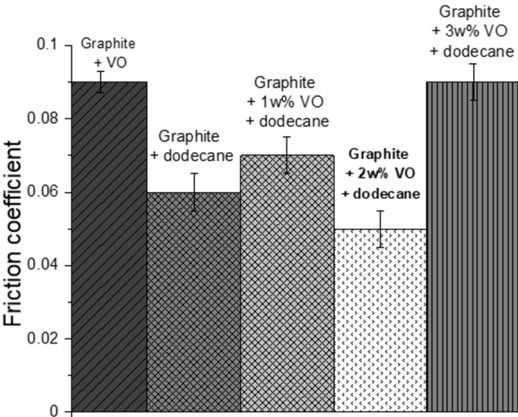

**Figure 4.** Friction coefficient values obtained at 2000 cycles of the mixtures containing graphite.

Moringa oil is mainly composed of unsaturated fatty acid molecules. Fatty acids are amphiphilic molecules constituted of a hydrophilic polar part (carboxylic acid) and a hydrophobic group (aliphatic chain). Hardy et al. (1922) showed that polar groups adhere on the steel surfaces in contact, and the fatty acid molecules orientate vertically to form close-packed monolayers [33]. Numerous studies in the literature are related to the influence of friction performances of vegetable oils as a function of their fatty acid [34–37]. Sharma et al. (2009) have shown that low amounts of saturated fatty acid and high amounts of unsaturated fatty acid result in low friction [18]. Reeves et al. (2015) have shown that natural oils with a high percentage of oleic acid preserve low friction coefficient values and low wear rates, because the oleic acid forms a denser and protective fatty acid monolayer that minimizes the asperity contact [30]. Bahari et al. (2018), studying the friction and wear responses of vegetable oil and their blends with mineral engine oils, showed that saturated acids exhibit a lower friction coefficient than unsaturated acids, linoleic and oleic acid [15]. All studies suggest that free fatty acids improve the lubrication properties of vegetable oils. Moreover, Crespo et al. (2018) worked on adsorption, self-organization and mechanical properties of different fatty acids layers under confinements states [10]. The molecule architecture of oleic acid presents one unsaturation, a double bond in cis configuration. These results in a bent shape for the alkyl chain compared with stearic acid in which alkyl chains are straight. On the base of our results, the thickness of the fatty acid molecules adsorbed on the surfaces appear as a key parameter to explain the friction properties of the lubricant. The adsorption of fatty acid molecules in the exfoliated graphite surface has especially to be investigated. It would be investigating to add different type of particles in VO/dodecane blends and compare the friction results. Nevertheless, the most important point to note is that the addition of 2 w% of moringa oil to conventional graphite/dodecane lubricant significantly improves the friction performances.

### 3.2. Physicochemical Characterization of the Best Mixture

3.2.1. VO/Dodecane Blends before the Friction Experiments

Physicochemical analyses of the blend with 2 w% of VO in dodecane without graphite particles have been investigated by FTIR and TGA. FTIR technique allows us to identify important functional groups in pure VO, which are capable to absorb metal ions. FTIR measurement uncertainty is approximately $\pm 3$ cm$^{-1}$. Figure 5 presents the FTIR spectrum of pure moringa oil with assignment of the different peaks [38]. Triglyceride is the major component in moringa oil. Triglyceride functional groups can be observed around 2937 cm$^{-1}$ (C–H stretching asymmetry), 2856 cm$^{-1}$ (C–H stretching symmetry), 1749 cm$^{-1}$ (C=O stretching), 1454 cm$^{-1}$ (C–H bending scissoring), 1166 cm$^{-1}$ (C–O stretching and C–H bending) and 709 cm$^{-1}$ (C–H bending rocking) [39,40]. FTIR peaks between 1400 and 1800 cm$^{-1}$ are attributed to C–H bending, C=O stretching and C=C stretching groups and are directly related to unsaturated C=C bonds: oleic and linoleic acids. In pure moringa

oil used, the intensity of C=C peaks is very small or negligible and consequently hardly detectable in the blend (Figure 6).

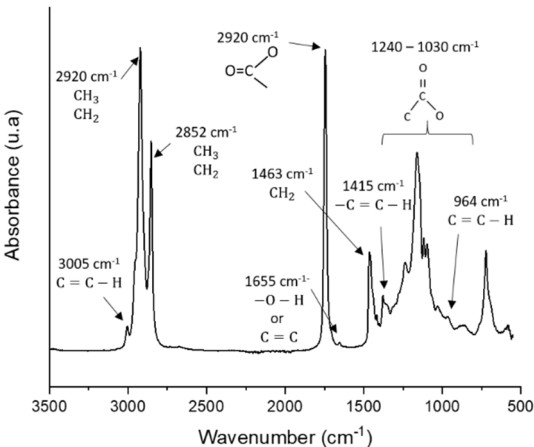

**Figure 5.** FTIR spectra of pure moringa oil.

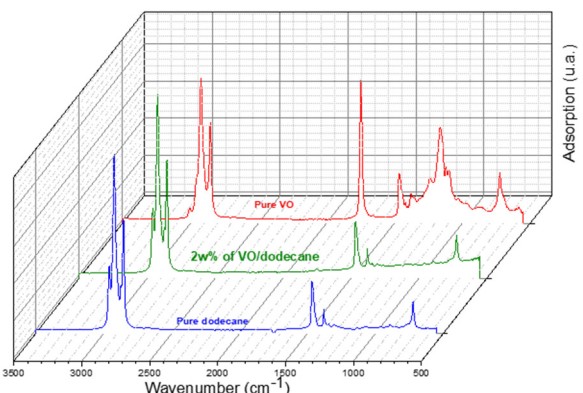

**Figure 6.** FTIR spectra of pure VO, pure dodecane and the blend of 2 w% of VO in dodecane.

Figure 6 shows a comparison between pure dodecane, pure VO and the blend containing 2 w% of VO in dodecane. Due to low amount of VO added, the carboxylic peaks characterizing fatty acid molecules have not been detected. FTIR spectra of the blend is similar to that of dodecane. The VO peaks are not detectable, but the presence of VO in dodecane have a beneficial action on the friction performances of the mixtures: $\mu_{2w\%VO+dodecane} \approx 0.1$ and $\mu_{Graphite+2w\%VO+dodecane} \approx 0.05$. The improvements are about 55% compared to performances of pure dodecane.

The influence of the presence of VO on the thermal stability properties of dodecane has been investigated by thermogravimetry (TGA). The analyses were carried out in an inert atmosphere of argon. Figure 7 displays the TGA curves for pure dodecane (Figure 7a), pure VO (Figure 7b) and the blend of 2 w% of VO in dodecane (Figure 7c). Both TGA and DTG curves reveal a high thermo-oxidative stability in pure VO. TGA curves confirmed by the DTG curve show three distinct stages of mass loss for pure VO. The first stage is associated with water desorption. From 30 °C to about 100 °C, only 2% of mass loss is observed for the samples. In the case of pure dodecane, the main thermal degradation takes place in a single continuous step with an onset temperature, $T_{onset-dodecane}$ of 150 °C. Dodecane boiling point is 216 °C. It vaporizes rapidly into the form of its gaseous species. For pure VO, two other stages are observed in the temperature range from 300 to 500 °C. These stages are related to the decomposition of the greater part of the oil components, which probably includes fatty acids. For example, stearic acid presents a boiling point of 383 °C. At 500 °C, no residue was observed.

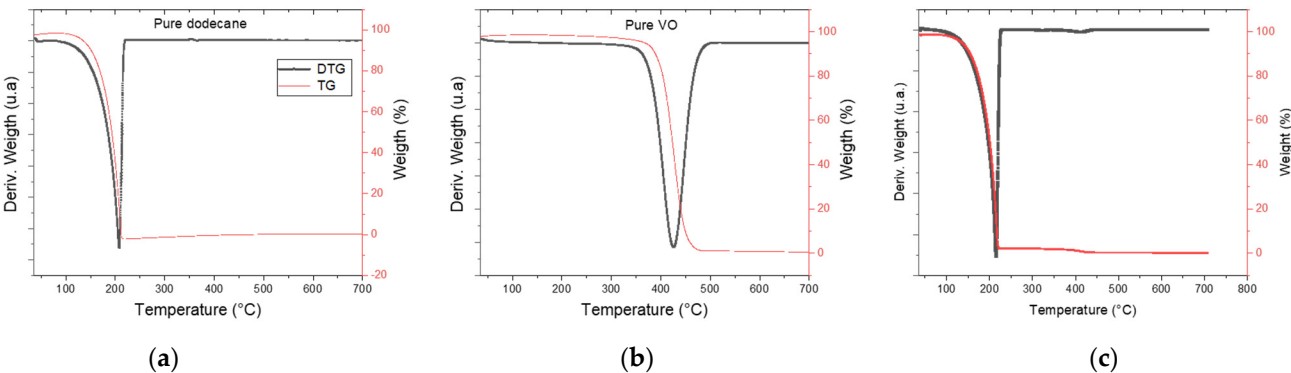

**Figure 7.** Thermo-degradation analysis of (**a**) pure dodecane, (**b**) pure VO, (**c**) 2 w% of VO/dodecane blend.

Vecchio et al. (2008) have investigated the thermal breakdown of triglycerides contained in olive oil [41]. They showed a single disintegration step between 160 and 370 °C on the TGA and DTG curves of saturated C18:0, whereas two overlapped steps occurred in the unsaturated chain. Different propositions were reported in the literature about the interpretation of the decomposition of vegetable oil in the temperature between 420 and 495 °C. Garcia et al. (2007) observed that the mass loss step is due to oxidation of unsaturated fatty acids in the 250 to 410 °C range, while it is attributed to the oxidation of the saturated fatty in the 410–480 °C range [42]. According to Santos et al. (2004), the polyunsaturated fatty acids decomposition should occur in the 200 to 380 °C range, then the monounsaturated fatty acids decomposition between 380 and 480 °C and, finally, the saturated fatty acids thermal decomposition in the range of 480–600 °C [43]. The comparison of the FTIR spectrum of moringa oil within the literature suggest that our VO is mainly composed of unsaturated fatty acid molecules. In the blend, a weak influence of the presence of VO is observed on the thermal degradation of dodecane. The first one is due to dodecane degradation. The two other superimposed steps of mass loss were observed between approximately 315 and 460 °C with very weak loss of mass (between 1 and 2%) corresponding to these thermal degradations of VO added.

3.2.2. Graphite/VO/Dodecane Blend after Friction Experiments

The obtained tribofilms have been investigated by SEM and Raman spectroscopy analyses. Figure 8 presents a SEM micrograph of the film formed with graphite/2 w% VO/dodecane blend. We can see that the wear trace is not homogeneous, characterizing weak adhesion with the steel plane. Some parts of the tribofilm are missing. The initial stripes are still visible on the steel plane, evidencing weak wear and, as a consequence, good antiwear properties of the lubricant.

Figure 9 displays a typical Raman spectrum recorded close to the tribofilm of the best mixture (Figure 9a) and another one recorded on the middle of the tribofilm (Figure 9b). The first one corresponds to the Raman spectrum of the initial graphite particles before friction. Both spectra exhibit the characteristic G, D and D' bands associated with the presence of graphite domains. The G band (1580 cm$^{-1}$) is attributed to the E$_{2g}$ vibration mode of the graphite lattice, while the D (1350 cm$^{-1}$) and D' (1620 cm$^{-1}$) bands are associated with disorder [44]. No significant difference is obtained indicating that the structure of graphite particles does not evolve during the friction test. In previous study, we have shown that in liquid conditions, the crystallographic disorder of graphite induced by the friction process is lower compared to the dry one [32]. In the presence of liquid (pure dodecane), the crystallites size decreases during the sliding experiments, but this reduction is less important than in dry conditions (pure graphite). In this study, the tribofilm investigations demonstrated that the presence of fatty acid thickness seems to attenuate the mechanical constraints limiting/avoiding the destruction of crystallites during the sliding process. In addition to liquid influence, demonstrated previously, an adsorption effect of fatty acid molecules has been evidenced on the tribological performances of graphite,

forming an adsorbed protective film on the graphite particles and steel surfaces during the tribological experiments.

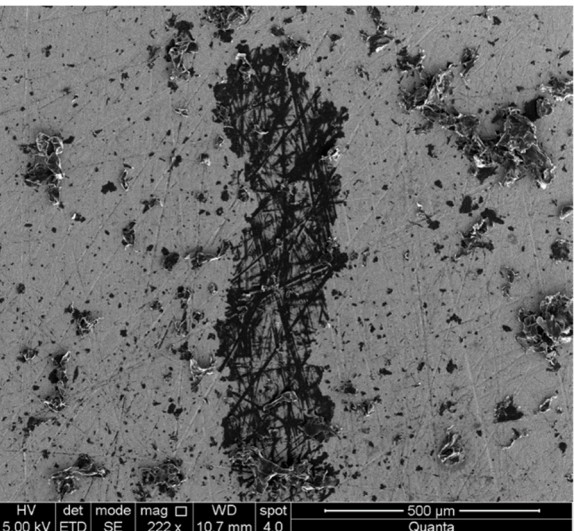

**Figure 8.** SEM picture of a tribofilm obtained during the friction experiment of the lubricant composed of graphite and 2 w% of VO in dodecane.

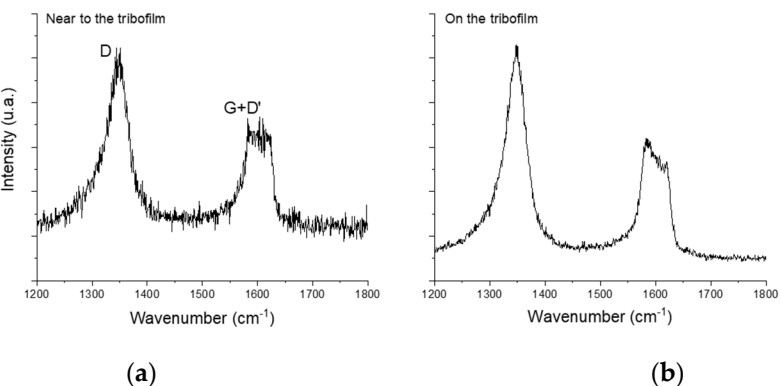

(**a**)                                                    (**b**)

**Figure 9.** Raman spectra recorded after friction experiment (**a**) near to the wear trace and (**b**) on the middle of the tribofilm.

## 4. Conclusions

Moringa oil presents excellent friction influence as bio-additive for lubrication. By studying the tribological performances of the different blends containing small amounts of VO in dodecane, an important reduction in the friction and wear performances of dodecane were identified. Dodecane was the synthetic base oil used. The friction reduction was about 55% in the presence of 1 w% of VO added. This improvement is attributed to the presence of fatty acid molecules. FTIR and TGA investigations of VO/dodecane blends lead to the conclusion that no significant influence related to the presence of moringa oil is observed on the physicochemical properties of dodecane. Despite the weak percentage of VO, the presence of fatty acid molecules improves the tribological performances of mineral-oil-based lubricant. A critical amount of VO and, consequently, a critical adsorbed thickness is necessary to ensure low friction performances. Moringa oil as a bio-additive also has an adsorption influence on the tribological performances of graphite particles. In the presence of different percentage of VO, the best results are obtained for a lubricant formulation containing 2 w% of VO. The hypothesis about the liquid effect is not enough to explain the friction reduction. Raman results and SEM investigations of tribofilms does not show significant change in graphite structure. Mainly constituted with unsaturated

fatty acid molecules, the presence of VO protects surfaces in contact by removing mechanical constraints due to sliding contact. These results show good antiwear properties of the lubricant.

**Author Contributions:** Conceptualization, N.N.-M.; methodology, N.N.-M.; software, Y.B. and P.B.; validation, N.N.-M. and P.T.; formal analysis, N.N.-M.; investigation; N.N.-M.; writing—original draft preparation, N.N.-M.; writing—review and editing, N.N.-M.; review, P.T., Y.B. and P.T.; project administration, P.T.; funding acquisition, P.T. All authors have read and agreed to the published version of the manuscript.

**Funding:** This research was funded by Région Guadeloupe.

**Acknowledgments:** Henry Joseph of Phytobokaz Laboratory for the vegetable oil used.

**Conflicts of Interest:** The authors declare no conflict of interest. The funders had no role in the design of the study; in the collection, analyses or interpretation of data; in the writing of the manuscript= or in the decision to publish the results.

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
