# Peer review of "Moringa and Graphite as Additives to Conventional Petroleum-Based Lubricants"

_lubricants, doi:10.3390/lubricants9070065_

Round 1

Reviewer 1 Report

Abstract

In line 20 of the text, it is important to define in advance the abbreviation VO. It is understood that it refers to vegetable oils. Therefore it would be important to define this abbreviation on line 13.

Introduction

Et alii, generally abbreviated as et al., Is a Latin phrase that literally means "and others." It is necessary that the expression et al. being written in a dead language, it is written in a cursive format.

In line 47, the bibliographic reference that mentions the research of Frye and collaborators is poorly described. It is necessary to indicate the year. Follow the indication shown at the top. This writing error is reproduced in the following lines: 51, Pereira; line 59, Kerni; line 69, Sue.

The sentences on line 59 through line 63 appear to be redundant. It is possible to restructure this section and condense it into a single clear and concise sentence.

Materials and Methods

It is important to point out the bibliographic reference from which the information contained in table 1 was obtained.

Check the font size of lines 119 to 129 of the materials and methods section. The font size is smaller than suggested by the journal template.

Results and Discussion

The sentence on lines 113-115 could be restructured as it begins by mentioning Figure 2, which has not been shown yet.

In the following lines of the manuscript the bibliographic reference that mentions the principal researcher and collaborators is poorly described. It is necessary to indicate the year. Follow the indication shown at section Introduction. In line 156, Bahari; in line 158, Reeves; in line 160, Fry; In line 194, Fry; in line 205, Hardy; in line 208, Sharma; in line 210, Reeves; in line 213, Bahari; in line 2016, Crespo; in line 272, Vecchio; in line 276, García; in line 279, Santos.

The sentence on lines 306-308 could be restructured as it begins by mentioning Figure 9, which has not been shown yet.

Conclusion

It is not clear how the analysis of the FTIR and TGA on the VO / dodecane mixtures provides evidence that the physicochemical properties of dodecane are not preponderant in these mixtures. It is necessary to enrich this discussion to improve the conclusion of the manuscript.

References

The 30% percent are references from the period 2016-2021, 27% from the period 2010-2015, 43% are references are are lower than the year 2010. In general, 70% of the references it is older than 5 years.

An effort has been made to update the references that support the research. Even so, it is recommended to have a greater number of references no greater than 5 years from the date the manuscript is submitted for review. The updated references allow us to observe the trends in the area of ​​lubricants and the novelty of the research described in the manuscript.

Author Response

I thanks you for your remarks. I send in attachment a track change for my point-by-point response.

Reviewer 2 Report

I reviewed a previous version and it seems OK

Author Response

I would like thanks you for wanting to review my paper.

This manuscript is a resubmission of an earlier submission. The following is a list of the peer review reports and author responses from that submission.

Round 1

Reviewer 1 Report

Dear authors

TITLE

The definition of green lubricants or biolubricants is a non-toxic, biodegradable substance from renewable sources added with an additive that has the same characteristics. In this case the dodecane that is the base of this lubricant is considered a mineral oil. It is important to consider that the title directly reflects the research described in this manuscript.

Abstract

In the abstract section it can be seen that the manuscript actually describes the development of a mineral lubricant comparing its performance with different mixtures of this lubricant with green additives such as moringa oil and graphite. This validates the view that the title of this manuscript is misfocused. It is necessary to reconsider that the title is consistent with the information described in the manuscript.

Introduction

The introduction is poor in describing current trends in studies of blends of mineral lubricants with green additives. It's not updated. The introduction can be improved by doing a bibliographic review directed to the study of mixtures of various vegetable oils in mixtures with mineral oil. Furthermore, it is necessary to study in depth the effects of graphite as an additive to improve the behavior of lubricant mixtures in tribological studies.

Materials and Methods

The experimental methodology is deficient when explaining the preparation of the mixtures, since it does not comment on how they were prepared, what type of gravimetric technique was used, the precision of the gravimetric method and the variance of the% w of the lubricant mixtures with additives

Furthermore, the temperature at which the tribological study is carried out is not explained. Since this is important to be able to evaluate the tribological performance of the lubricant and compare on equal terms with the other lubricant + additive mixtures.

The reason for the infrared study and what the research group hopes to observe with this analysis is not clearly defined in the methodology. It is important to detail the infrared analysis methodology focused on the objective of this study for mineral lubricant + additive mixtures.

Results and Discussion

This section generally discusses the tribological behavior of lubricant + additive mixtures, but no new information is provided on their behavior. For example, the reason for the behavior of the graphite + 3% W VO + dodecane mixture is not adequately described. It is only mentioned literally that it is the one that does not behave in a normal way to the other mixes. Furthermore, the literature on which the discussion of these mixtures is based is not up to date. Review new literature and improve discussion of tribological study results.

The FTIR study is not clearly delimited since the actual composition of moringa oil is not detailed in the experimental methodology and it is only assumed that it is pure vegetable oil only composed of triglycerides. But nowhere is the pretreatment process described to warrant this assumption.

Furthermore, it is important to note that the FTIR spectra of the mixtures do not show the presence of fatty acids, which is attributed to the presence of graphite in the mixtures and the low concentration of vegetable oil in the mixture. Therefore, it is important to improve the discussion of the results of this section by updating the references that serve as the theoretical framework of the manuscript.

In the discussion of the TGA result, it remains to be explained whether it was carried out in a normal air mixture or in an atmosphere of pure nitrogen. This is important to define in the experimental methodology to be able to substantiate the observations that the TGA provides.

Conclusion

The conclusions do not provide new information on the use of biodegradable additives in mineral oils. These conclusions could be improved if they are developed based on new advances in the study of biodegradable additives. The study is not conclusive on the presence of free fatty acids in the mixtures or the presence of certain types of fatty acids that make up triglyceride. Which of these two chemical characteristics of the mixtures impact their tribological behavior. This confuses the conclusions reached by this study. It is recommended to improve the focus of the conclusions based on current research.

References

Bibliographic references are not up to date, 30% percent are references from the period 2015-2021, 15% from the period 2010-2014, 33% are references from the period 2000 to 2009 and 22% are prior to the year 2000. In general, 70% of the references It is older than 5 years. Therefore, it is recommended to update the bibliographic research to review the novelties that have been presented in the context of biolubricants.

Reviewer 2 Report

Please. make a better paper. Now is in Major level.
